# Modern Microbiological Methods to Detect Biofilm Formation in Orthopedy and Suggestions for Antibiotic Therapy, with Particular Emphasis on Prosthetic Joint Infection (PJI)

**DOI:** 10.3390/microorganisms12061198

**Published:** 2024-06-14

**Authors:** Paweł Mikziński, Karolina Kraus, Jarosław Widelski, Emil Paluch

**Affiliations:** 1Faculty of Medicine, Wroclaw Medical University, Wyb. Pasteura 1, 50-376 Wroclaw, Poland; pawel.mikzinski@student.umw.edu.pl (P.M.); karolina.kraus@student.umw.edu.pl (K.K.); 2Department of Pharmacognosy with Medicinal Plants Garden, Lublin Medical University, 20-093 Lublin, Poland; jaroslaw.widelski@umlub.pl; 3Department of Microbiology, Faculty of Medicine, Wroclaw Medical University, Tytusa Chalubinskiego 4, 50-376 Wroclaw, Poland

**Keywords:** biofilm, orthopedic surgery, endoprostheses, postoperative infections, anti-microbial treatment, biofilm detection methods

## Abstract

Biofilm formation is a serious problem that relatively often causes complications in orthopedic surgery. Biofilm-forming pathogens invade implanted foreign bodies and surrounding tissues. Such a condition, if not limited at the appropriate time, often requires reoperation. This can be partially prevented by selecting an appropriate prosthesis material that prevents the development of biofilm. There are many modern techniques available to detect the formed biofilm. By applying them we can identify and visualize biofilm-forming microorganisms. The most common etiological factors associated with biofilms in orthopedics are: *Staphylococcus aureus*, coagulase-negative Staphylococci (CoNS), and *Enterococcus* spp., whereas Gram-negative bacilli and *Candida* spp. also deserve attention. It seems crucial, for therapeutic success, to eradicate the microorganisms able to form biofilm after the implantation of endoprostheses. Planning the effective targeted antimicrobial treatment of postoperative infections requires accurate identification of the microorganism responsible for the complications of the procedure. The modern microbiological testing techniques described in this article show the diagnostic options that can be followed to enable the implementation of effective treatment.

## 1. Introduction

In modern orthopedic surgery, biomedical implants (such as screws, nails, etc.) are commonly used in treatment. Endoprostheses have also become a regular treatment for advanced osteoarthritis, with orthopedic surgeons in France performing 124,251 total hip replacements in 2018 [1]. Such interventions come with a risk of a biofilm-related infection, because applied foreign bodies are prone to biofilm-forming microorganisms’ colonization [2,3]. Biofilm formed by microorganisms causes a multifold increase in treatment costs, with a much higher risk of failure for antimicrobial therapies [4]. Treatment, in terms of the above-mentioned infections, can be very complex and often includes surgical removal of the infected prostheses, prolonged antibiotics admission and revision surgeries. The above-mentioned factors show how important it is to detect those complications as soon as possible [5,6]. In terms of ways of detecting biofilm infections, most hospitals use cheap, classic technologies, such as culture methods. These methods have been used throughout the medical systems for many years and they work very well when it comes to many acute epidemic diseases, but they are not as efficient in biofilm-related infections. In many medical cases, including septic arthritis, cultures give false-negative results [7]. False-negative culture results can lead to the progression of early stage infections to more dangerous and serious infections, and even to the development of sepsis. In this case, modern methods of diagnosing biofilm formation can really help [8]. 

In our review, we will discuss the classic as well as modern methods of diagnosing orthopedic biofilm-related infections, and the current antimicrobial therapy applied in those types of infections.

## 2. Materials and Methods

We collected data on the current novel diagnostics and the treatment of biofilm, in particular PJIs. We used keywords such as “PJI novel diagnostic”, “biofilm detection microbiological methods”, ”biofilm formation”, “PJI biofilm treatment” “PJI antibiotics”. We analyzed the methods in terms of usefulness, especially in orthopedics. We have also compiled the other most popular and innovative PJI detection methods available based on current data from available reviews. Additionally, based on guidelines and data from various centers, we have summarized the currently used treatment methods. Whenever possible, we used sources dating back no more than 10 years.

## 3. Biofilm

### 3.1. Biofilm Formation

A biofilm is usually a multi-species spatial structure of microorganisms that pro-vides them with many benefits, including increased resistance to many unfavorable conditions like desiccation and starvation. What is important from a clinical point of view, is that this form, combined with the fraction of cells with reduced metabolic activity (persisted cells), makes the biofilm structure more resistant to antibiotics [9]. Biofilm is a collective of microbial cells; it usually begins with a few cells which adhere to the surface and divide many times and form a polymer matrix, which later matures and creates more advanced microcolony systems and a complicated spatial architecture of mature biofilm (often with water channels), until the dispersion of planktonic forms [10]. The most common infection-causing microorganisms associated with biofilms in orthopedics are: coagulase-positive and coagulase-negative staphylococci (CoPS and CoNS), *Enterococcus faecalis*, Gram-negative rods and *Candida albicans*. The *Candida* genus is the most usual cause of fungal periprosthetic joint infections. Figure 1 presents the spatial structure of the *C. albicans* biofilm formed on a metal surface. *Cryptococcus neoformans*, *Trichosporon* spp. and *Aspergillus fumigatus* are other biofilm-forming fungi that can cause infections in humans [11]. This form of growth, similar to bacterial biofilms, provides them with more nutrients, metabolic cooperation, defense, and the exchange of many unique capabilities. This form also makes them much more resistant to antifungal therapy than planktonic forms [12,13]. Biofilm as a structure plays a crucial role in survival in unfavorable conditions of the external environment; however, when it also forms on the surface of human body (implant surface), it can cause serious infections, even leading to death.

### 3.2. Biofilm Formation in Static (Non-Flow) and Flow Conditions

Recently, more and more research concerns not only the study of biofilm in static conditions, but also in microfluidic conditions, i.e., those that are similar to those in our body. At the same time, in the above topic, it must be assumed that this type of biofilm may be formed in certain conditions, e.g., inside vascular catheters after orthopedic procedures (implant-related infections) and in highly vascularized places. In this way, the infection can spread very quickly throughout the body via the vascular route [14,15].

The initial stages of biofilm formation, if we mention the adhesion of microorganisms to surfaces, may be very similar for both conditions. This process begins with an initial attachment to a certain surface (biotic or abiotic) which eventually becomes irreversible. Then, the production of extracellular polymeric substances (EPS) that consist of polysaccharides, lipids, proteins, and nucleic acids starts [14]. After the attachment microcolony formation begins, with some bacterial species naturally unable to attach to the surface connecting themselves to the cells already attached, this activity contributes to the matrix creation [16]. Although the processes occur in a similar way for both conditions (static and in flow condition), the difference occurs over time. For flow models, we can often observe cell adhesion from the first hour (0–2 h), which for a static model (0–6 h) may often go unnoticed due to the formation of a sediment of microorganisms, even under shaking conditions (although it may physically occur). Moreover, in the flow model, a shear current appears, which often hinders adhesion and causes microcolonies to form in the form of flat but elongated “streaks”. The next step is the maturing period of the created biofilm. The created stable conditions enable free gene transfer and further EPS formation. It is worth mentioning the molecular language of microbial communication, which is Quorum Sensing (QS), which controls many metabolic processes of the microorganism, including the formation of biofilm, which many antimicrobial compounds, including antibiotics, can affect by blocking it. This may happen at various stages of biofilm formation [3].

Interestingly, the dynamics of biofilm growth are much higher for flowing conditions than for static conditions (skipping shaking). The slow increase in the adhesion clusters was observed for microfluidic conditions from 2 to 8 h and in static even up to 18 h. The real dynamics of biofilm growth are observed even after 8–18 h in flow conditions, which can be explained by the greater abundance of nutrients. In the static condition, a similar biofilm area on surfaces can often be observed only after 18–48 h, depending on the microorganism. Moreover, specific biofilm balls often appear in flow conditions (biofilm fragments break off and flow further, creating further infection foci). In case of fungi, this phenomenon was called “fungus ball”, which does not occur in static conditions. In extreme cases, the biofilm may develop so quickly that within 18–24 h it may cause an almost complete clogging of the flow channel, e.g., an intravascular catheter. Whereas, in static conditions, a mature spatial biofilm is often formed within 48–72 h [11,15,17].

The above description shows the difference between static models and the modern flow model of biofilm formation, which better reflects the conditions that occur in our bodies (although both models can occur under appropriate conditions). However, all this creates an advanced microbial community, which provides the bacteria and fungus with a perfect environment to exchange antibiotic-resistant genes, create nutrient gradients, and generally survive many adverse circumstances [15,18]. Biofilm formation processes in static conditions—standard model (Figure 2A—top) and biofilm formation in microfluidic conditions (Figure 2B—bottom).

### 3.3. Biofilm Related Infections in Orthopedy

Biofilm-forming bacteria are involved in many infections, affecting different fields of medicine [15]. They usually form on a non-living surface; therefore, they are a big issue in orthopedy where biomedical implants, endoprostheses, are commonly used. We can divide those complications into chronic bone and marrow infections (osteomyelitis), implant-associated infections, and those that combine both of them [19]. In the cases of bone infections, biofilm creates a lot of problems in chronic osteomyelitis. As its attachment to the inorganic bone structure and surrounding tissues becomes irreversible, the treatment requires surgical debridement, in addition to the antibiotic therapy [20]. Infections associated with endoprostheses are usually referred to as prosthetic joint infections (PJI). Currently, there are a few proposed criteria defining the presence of PJIs. The Musculoskeletal Infection Society (MSIS), in 2018, proposed the most commonly used concept [21].

### 3.4. Periprosthetic Joint Infections’ Etiology

The etiology of PJIs can vary according to what type of PJI is presented in each case. The most common bacterial species in all types of PJIs is *S. aureus* [22]. In early PJIs, the most common pathogens are *S. aureus*, aerobic Gram-negative bacilli, beta-hemolytic streptococci, and *Enterococcus* spp. When it comes to delayed PJIs, they are usually caused by coagulase-negative staphylococci, *Cutibacterium acnes*, *Enterococcus* spp. and *Staphylococcus aureus*. Whereas in late infections, we usually deal with *S. aureus*, coagulase-negative staphylococci, streptococci of viridans group, enterococci, and occasionally Gram-negative bacilli [23]. Only 1% of PJIs are fungal infections. The most common species responsible for those are *Candida* spp. (with the most common being *C. albicans*, *C. parapsilosis* and *C. glabrata*). The other fungi involved in PJIs, although they occur very rarely, are *Aspergillus* spp., *Pithomyces* spp., *Penicillium* spp., *Rhodotorula minuta*, *Virticillium* spp., *Blastoschizomyces capitatus*, *Alternaria* spp., *Trichosporon* spp. and *Aureobasidium* spp. [24]. For the purposes of this work, a graphic illustrating the basic structure of the knee prosthesis and biofilm structure of *C. albicans* on a titanium implant was created (Figure 3).

## 4. Implant against Biofilm Materials

Once attached to the implant material, bacteria begin biofilm formation. The period from the initial adhesion of bacterial cells to the point of irreversible binding and early biofilm formation is seen as a potential therapeutic ‘window of opportunity’ to eliminate PJI before a biofilm is fully established. During this phase, bacteria remain vulnerable to conventional antibiotics and immune system attacks [25]. There are two main approaches to improving the materials used for prostheses: preventing biofilm adhesion—passive surface finishing/modification (PSM), eradicating microorganisms—active surface finishing/modification (ASM). The third way, however, not directly interacting with the implant itself, are local carriers or coatings (LCC)—applied during a surgical procedure, either immediately before or at the same time as the implant, and around it [26]. All widely used orthopedic materials, including cobalt-chromium, titanium, polyethylene, polymethyl methacrylate (PMMA), and ceramics, are prone to colonization by biofilm-forming bacteria [27]. Sorrentino et al. showed in in vitro studies that ceramics possess beneficial physicochemical surface properties that make them less conducive to biofilm formation compared to other implant materials, showing reduced bacterial adhesion and slower biofilm development (*Staphylococcus aureus* and *Staphylococcus epidermidis* biofilm were used) [28].

### 4.1. Passive Surface Finishing/Modification (PSM)

These surfaces are designed not to release bactericidal agents into the surrounding tissues, but to prevent or reduce bacterial adhesion by modifying their surface chemistry and/or structure. Surface features are crucial factors influencing bacterial attachment and the subsequent development of biofilms [26]. One of the possibilities for improving the material is to select its appropriate texture. Paulitsch-Fuchs et al., in their study, compared this factor on the example of a cobalt–chromium–molybdenum implant surface. They showed that CoCrMo surfaces, with increased roughness due to modifications, are more prone to biofilm formation, exhibiting elevated levels of proteins and polysaccharides [29]. Nanostructured titanium surfaces refer to titanium surfaces that have been engineered or modified at the nanoscale level, typically with features such as nanotubes, nanoparticles, or nanograins. These surfaces offer unique properties, such as increased surface area, enhanced mechanical strength, improved biocompatibility, and potentially reduced bacterial adhesion. Cao et al. compared the adherence of *S. epidermidis* to three types of titanium surface: polished, spear-type and pocket-type. They found that the spear-type nanostructure was significantly more effective in preventing bacterial adhesion than the pocket-type nanostructure. However, in terms of bactericidal efficiency, the spear-type nanostructure performed slightly less effectively than the pocket-type [30]. Another method of modification is to cover the implant with various substances. Coating titanium implants with polymer coatings, such as hydrophilic polymethacrylic acid, polyethylene oxide, or protein-resistant polyethylene glycol, effectively reduces bacterial adhesion. [26]. Hydrophobic and super-hydrophobic surface treatment technologies have demonstrated a significant antibacterial repellent effect in preclinical studies [31]. Another possible coating is hydrogel coating, with can be improved and used as an antifouling, antibacterial, drug-loaded and light-responsive type [32]. Rivado et al. characterized the promising strategy of using the anti-adhesion activity of biosurfactants produced by *Bacillis* spp. However, testing biosurfactants have not been found to be effective against a wide range of bacteria [33].

### 4.2. Active Surface Finishing/Modification (ASM)

ASM can be divided into main types: inorganic (with nanoparticles, metal ions, iodine, non-metal ions), organic (with antibiotics, antimicrobial peptides, cytokines, chitosan, enzymes and biofilm-disrupting agents) and synthetic (with non-antibiotic antimicrobial compounds). Currently, those with silver ions, nanoparticles, and those coated/linked with antibiotics are used practically and on a wider scale [34].

Silver coating is the most widely used metal coating, because dissolved silver cations are biochemically active; they disrupt bacterial cell membrane permeability, interfere with bacterial metabolism, and contribute to the formation of reactive oxygen species [35]. A silver ion-doped hydroxyapatite coating used in an in vitro study conducted by Sevencan et al. demonstrated better outcomes than only a hydroxyapatite coating and uncoated titanium [36]. The novel solution is to use doping with Sr^2+^ hydroxyapatite as a drug carrier. Tetracycline was selected as the model drug in a study by Tsai et al., which proved that such a coating ensures good availability and optimal drug concentration [37].

In a clinical study, the effectiveness of iodine-coated titanium megaprostheses was demonstrated, resulting in excellent outcomes [38].

Graphene-based materials exhibit potent antimicrobial properties and effectively inhibit bacterial colonization. A graphene coating on titanium can reduce the formation of biofilm, because of the hydrophobicity of this allotrope of carbon [39].

Antibiotics need delivery vehicles to be used locally in prothesis. A potential drug carrier in joint prothesis is polymethylmethacrylate (PMMA), currently used in orthopedy [40]. The use of bioactive glass and mesoporous bioactive glass as a carrier of antibiotics, which has antibacterial properties in itself, seems to be an interesting concept [41]. Romanò et al. showed that the application of antibiotic-loaded hydrogel coatings on hip and knee prosthetic implants leads to a decrease in the incidence of early surgical site infections, with no observed adverse effects [42].

Chitosan possesses promising attributes as a biomaterial, including biocompatibility, biodegradability, minimal immunogenicity, antimicrobial properties, and potential for reducing inflammation [43]. Some research suggests that chitosan derivatives can be securely attached to titanium alloys and exhibit protective effects against certain bacterial species, either independently or when combined with other antimicrobial agents, such as vancomycin [44]. 

## 5. Microbiological Methods of Diagnosing Biofilm Formation in Orthopedics

### 5.1. Direct Microbiological Culturing

Microbiological culturing is the most common method in diagnosing periprosthetic joint infections. The materials used for such cultures are either synovial fluid or tissue samples that were obtained intra-operatively. Usually, three to five tissue samples are taken during surgery; however, according to studies, obtaining five samples and growing them for at least 8 days provides better results [45]. According to a recommendation from a national periprosthetic joint infection sampling and culture guide from 2018, the best combination for culturing samples is a blood culture bottle (5 days), a chocolate agar plate (7 days) and anaerobic broth (14 days). In order to recognize mixed cultures, selective agar should be applied as in: Columbia CNA Agar with 5% Sheep Blood (CAN), MacConkey Agar (MCA) and selective anaerobic plates—Schaedler Anaerobe Agar (SAA)/Thioglycollate Bulion (TB) for broth subcultures when the direct cultures have already grown staphylococci or Gram-negative bacilli [46]. This method is definitely the most obtainable one in most hospitals and was considered a gold standard in PJI cases, but unfortunately a huge issue is caused by culture-negative periprosthetic joint infections.

### 5.2. Culture-Negative PJIs

The main problem with the direct culturing of intra-operatively obtained samples is the risk of culture results being false-negative [47]. Culture-negative PJIs occur in 20 to 50% of the patients that have clear indications of PJI presence. The usual cause of false-negative results is antibiotic treatment prior to sample obtaining. Most patients do not require antibiotic therapy before revision surgery, with the exceptions being critically ill cases and patients with septic symptoms [48]. According to the American Academy of Orthopedic Surgeons, if the patient received antibiotics, they should be discontinued for at least 2 weeks before the surgery to obtain reliable intra-operative samples [49]. Another cause for negative cultures are certain species that require specific conditions or more time. According to studies, such PJIs cases are often caused mycobacteria (43%), fungi (46%), *Cutibacterium acnes*, *Listeria monocytogenes*, *Brucella*, *Coxiella burnetii*, and other rarer organisms (11%). Another reason for the unreliable results are laboratory factors, such as inappropriate specimen collection and transportation, inaccurate medium or time for growth, and the contamination of sample [50,51]. There are some ways to improve the culturing method and minimize the risk of obtaining false-negative results. 

### 5.3. Culturing Sonication Fluid

Sonication is a method that uses low-intensity ultrasound to break down the biofilm formed on extracted implants. The explanted material is placed in a container with 400 mL of Ringer solution. The next step is the series of vortexing (30 s) and sonication (5 min), which is then repeated in the same order. The sonification fluid, obtained thanks to the above-mentioned processes, can then be used for culturing. Sonication fluid cultures are usually incubated up to 14 days [52,53] The next part of the process is of great importance because it determines whether the culture result is positive or negative. This information can be obtained using the CFU (colony-forming unit). According to Piper et al., the cutoff value for the CFU to determine the clinically significant result of cultured sonication fluid should be ≥20 CFU per plate [54]. Sonication fluid can be useful not only for culturing but also when applying molecular diagnostic methods for PJI, such as PCR. In terms of PCR, incorporating sonication fluid as an additional sample alongside intra-operative tissues can enhance its diagnostic value and provide valuable information in detecting PJI. [55] Synovial fluid can also be applied for biomarkers analysis, for example: alpha-defensin, leukocyte esterase, synovial fluid C-reactive protein, etc. Those common biomarkers come with high specificity and sensitivity, offering valuable information in PJI diagnosis [56]. According to Portillo et al., inoculating sonication fluid into blood culture bottles can enhance its diagnostic value. Their 2015 study utilized aerobic and anaerobic BacT/Alert FAN blood culture bottles with antimicrobial removal systems and incubated them following the inoculation of synovial fluid in the automated BacT/Alert system. Its efficiency was compared to standard sonication fluid cultures and intra-operative tissue sample cultures. The results indicated that this method can significantly improve the effectiveness of synovial fluid in diagnosing PJIs, particularly in cases involving patients who had previously received antimicrobial therapy. [57]. When it comes to comparing standard tissue sample cultures and sonication cultures, according to Trampuz et al., sonication fluid cultures offer superior sensitivity (78.5%) to standard tissue culture (60.8%). It also comes with similar sensitivity to synovial fluid culture. In terms of specificity, all three of those methods are comparable (sonication fluid culture—98.8%, standard tissue samples culture—99.2% and synovial fluid culture—98.1%). Additionally, sonication fluid cultures are also more sensitive than standard tissue cultures when dealing with patients who were under previous antimicrobial therapy [58].

### 5.4. DTT Pre-Treatment Method

D-L dithiothreitol (DTT) is a sulfhydryl compound capable of disrupting disulfide bonds between polysaccharides and neighboring proteins. This property enables it to serve as an antibiofilm agent, without posing any harmful effects to bacteria when administered at appropriate concentrations and durations. These capabilities have positioned this substance as potentially beneficial in addressing culture-negative PJIs. In a study by Drago et al., DTT was applied to synovial fluid samples collected from patients categorized as infected or non-infected using the MSIS criteria. These samples were then compared with synovial fluid samples that had not been pre-treated with DTT. The results demonstrated that applying DTT pre-treatment was superior to using standard samples. With DTT pre-treatment, the sensitivity of microbiological examination increased from 54.3% to 77.1% compared to standard samples, with no significant change in specificity [59]. A similar study, also by Drago et al., compared the DTT method with sonification. Both methods were applied on prosthetic joint materials explanted from patients (with aseptic loosening and presenting typical PJI symptoms). The results showed that pre-treatment with DTT can produce an equivalent or even better output to the sonication method. This study also observed that the DTT method possesses particularly high sensitivity towards *Staphylococcus epidermidis*, a common cause of PJIs. Its biggest advantages over sonication are the lowered risk of contamination and the fact that it does not require specialized equipment. Although it presents promising results, more studies are needed to validate its practical utility [60]. Another study by Sambri et al., showed similar results, ranking the DTT pre-treatment method and the sonication method as no different in sensitivity. It also presented them as superior to the standard culture method [61].

### 5.5. BioTimer Assay (BTA)

Another alternative method which can be potentially applied in diagnosing PJIS is BioTimer Assay (BTA). The BioTimer Assay indirectly identifies microorganisms by detecting their metabolic products, utilizing a unique reagent with indicators such as Phenol Red or Resazurin. The BTA is incapable of identifying microbial species. [62]. The BTA cannot identify microbial species but can be considered as a valuable tool for analyzing the antibiotic susceptibility of *Staphylococcus* spp. in biofilms [63].

### 5.6. Agar Encasement Culturing Method (AECM) with Candle Dip Method

AECM is a novel method, during which whole explanted materials are submerged in molten agar. This technique allows biofilm to grow without being disrupted. Moreover, it allows for the growing of species that are oxygen-sensitive. Unfortunately, this method’s limitation is the difficulty to visualize growing colonies, as a result of the non-transparency of the agar coating. A study by Moley et al. proposed the candle dip method to overcome AECM limitations. During this technique, explanted components were rinsed twice with phosphate-buffered saline (PBS) in order to get rid of planktonic bacteria and tissue or blood. After rinsing with PBS, the materials were placed in sterile containers, where agar coating took place. All components placed in the sterile environment were coated with molten agar in—1.5% Brain Heart Infusion (BHI) Agar or Typic Soy Agar (THI). After coating, components should have a thin layer of around 1 cm of agar on them. The next step was the incubation process. According to this study, incubation should take place in a sterile container under filtration units, and last for up to 12 days at 37 °C with 5% CO_2_. During this process, the agar coating has to be refilled after every 24 h period in order to not dry it out [64]. In this method, while the incubation process takes place, components are observed for any signs of growing colonies. If such a colony appears, its localization is noted and pictured, providing information about what parts of endoprostheses are prone to biofilm formation. What is more, a colony can be selected and sterilely collected for further analysis. In the study cited above, the candle dip method demonstrated a sensitivity rate of 60% and a specificity rate of 100% when compared with the MSIS criteria and culture status.

## 6. Molecular Diagnostic Methods

### 6.1. PCR

Molecular techniques can be very helpful in the management of culture-negative periprosthetic joint infections. One of the most common techniques to use in PJI diagnosing is Polymerase Chain Reaction (PCR). PCR can amplify a single or a few copies of a fragment of DNA [65]. In the context of periprosthetic joint infections, PCR entails extracting samples from the infected joint, like joint fluid or an intra-operative tissue biopsy (the same materials are used for direct culturing). The samples, after sonication, can also be applied for better results. Subsequently, PCR is employed to magnify and identify the DNA or RNA of the particular pathogens potentially responsible for the infection. In PJIs, usually 16S rDNA is the target [66].

### 6.2. Specific PCR

This method detects solitary bacterial species or related bacterial groups. It is exceptionally perceptive when it comes to the detection of the previously specified organisms of interest. In the case of PJIs, usually multiple species are targeted, so broad-range PCR is a more sufficient technique, although specific PCR can be useful if a likely cause of periprosthetic joint infection is known [67]. A study from 2022 by Prinz et al. applied sensitive multiplex RT-qPCR designed to detect *Cutibacterium* using sonication fluid explanted from patients with PJI as a sample. *Cutibacterium* PJIs are a huge problem because of their often asymptomatic nature and the need of a prolonged cultivation time. The results proved that this multiplex qPCR can potentially accelerate the time of diagnosing such cases, although more studies are needed in order to prove it [68]. Another study by Como et al. showed that applying qRT-PCR may be effective in identifying *C. acnes* infections, but it also possesses a risk of obtaining false-positive results [69]. 

### 6.3. Broad-Range PCR

This technique is more optimal for PJIs than specific PCR, which detects only single bacterial species or a closely related group, whereas in this case multiple species are targeted. This technique amplifies a diverse array of DNA sequences without focusing on individual genes or organisms [70,71]. Unlike using primers tailored for specific genes or species, broad-range PCR employs primers that are relatively uniform across various organisms (most of the broad-range PCR methods are grounded on a subunit of the bacterial ribosome—16S rDNA), enabling the amplification of a wide spectrum of DNA sequences [72]. Following amplification, the DNA products can undergo sequencing and then are compared to databases to determine the organisms existing within the sample. Laboratories may create their own self-made methods to perform PCR analysis in PJI cases. There are also many commercial PCR kits designed for the diagnosis of bone and joint infections, such as Unyvero i60^®^ multiplex PCR [73]. With this method, most of the microorganisms causing PJI can be detected even in small volumes in the range of few hours. Nevertheless, it has some limitations. In terms of PCR, hospitals would require prepared laboratories, specialized personnel and sufficient sample collection, which is not possible in many cases due to its costs. Another disadvantage is the risk of contamination. 

### 6.4. Next Generation Sequencing (NGS)

Next generation sequencing (NGS) rapidly sequences DNA or RNA molecules in high volumes, offering detailed insights into the genetic material after analysis. It simultaneously sequences millions to billions of fragments, aiding in various applications, such as pathogen detection, antimicrobial resistance profiling, and epidemiological surveillance [74]. Similarly to PCR, the focal point of bacterial identification is the 16S rRNA gene. Studies involving this method show some differences between them in terms of its sensitivity and specificity. There is more work needed in some fields in which this technique is applied, one of them being the thresholds for the single bacterial species percentage. One of the disadvantages of NGS is the risk of contamination and receiving false-positive results. Although this method shows a lot of potential in negative-culture PJIs, diagnosing it still requires further tests to take place. 

### 6.5. Metagenomic Next Generation Sequencing (mNGS)

For pathogen detection, the NGS-based metagenomics approaches can be useful sequencing techniques. Metagenomics next generation sequencing (mNGS) is a technique which sequences genetic material directly from environmental samples or clinical specimens without the need for prior culturing [75]. This method, like PCR, can potentially be applied in culture-negative PJIs, and according to Mei et al. article it is more efficient in detecting pathogens than PCR and culture methods [76].

### 6.6. Fluorescence In Situ Hybridization (FISH)

Fluorescent in situ hybridization (FISH) is a cytogenetic method which detects specific sequences (e.g, rRNA, mRNA, DNA) through the use of specific fluorescent DNA probes [77]. When imaging biofilms, a specific fluorescently labeled DNA marker is hybridized with its target inside tissues or cells. For bacterial biofilms, 16S/23S and 18S/28S rRNA sequences are targeted. The whole process consists of four parts. Firstly, sample cells are fixated and the cell membrane is permeabilized using chemical fixatives, which enhances efficiency. The next step is the hybridization with the target marker in controlled conditions. This step usually involves the denaturation of the sample DNA using 70% formamide2XSSC solution at 68–70 °C for 2 min to allow access to the target sequences [78]. After that, the excess probe is washed away to reduce background fluorescence, and finally visualization can be performed. To visualize samples, epifluorescence microscopy or confocal laser microscopy (CLSM) are used (Figure 4) [79].

An effective way to visualize biofilms is to combine the CLSM method with FISH. A study from 2014 applied multiplex fluorescence in situ hybridization (M-FISH, used to study multispecies biofilms) combined with CLSM to analyze oral biofilms. This combination was used to analyze early and late oral biofilms involving the following species: *Streptococcus* spp., *Fusobacterium nucleatum*, *Actinomyces naeslundi*, *Veilonella* spp. A five color M-FISH test was visualized with the use of CLSM, resulting in high-resolution images. Later, with a suitable analysis program, image processing and data analysis provided the quantification of various targets involved in those oral biofilms [80]. This shows how this effective combination can be used to study different bacterial biofilms, also those involved in PJIs.

## 7. Microbiological Methods of Visualizing Biofilms

### 7.1. Scanning Electron Microscopy

Scanning electron microscopy is a powerful imagining technique. It uses a focused electron beam that is swept across the specimen’s surface. As the electrons interact with the atoms in the sample, multiple signals are emitted. Detectors collect those emitted signals, allowing this tool to provide a detailed, three-dimensional image of the sample’s surface. After choosing a specimen’s sample for the examination, it has to be chemically fixated (Sabatini’s method variations) to preserve the structure of all cells and tissues. The next step is the dehydration process, and later the attachment and orientation of the sample [81,82]. This method has proven to be useful in visualizing biofilms in clinically infected implants. A study performed by University of Southampton used SEM to image and confirm the presence of biofilm on craniofacial osteosynthesis plates applied in mandibular fracture treatment [83]. In another study from 2021, SEM was successfully applied on orthopedic implants in order to visualize biofilm formation and learn its preferable location on implant parts [84].

### 7.2. Confocal Laser Scanning Microscopy (CLSM)

Confocal laser scanning microscopy is a very useful tool in studying biofilms. It can be promising when it comes to orthopedic biofilm related infections, especially when combined with fluorescent in situ hybridization (FISH) [85]. Confocal laser scanning microscopy (CLSM) was introduced in the late 1980s. It is a specialized form of fluorescent microscopy that increases the optical resolution by using a pinhole aperture to block out-of-focus light within the image. Its advantage over a conventional epifluorescence microscope is its ability to capture high quality digital images (also in the Z axis) of fluorescently labeled tissue. Before its introduction, it was impossible to obtain such pictures, because the signal from the tissue of interest would get overwhelmed by the background [86]. CLSM uses a laser that generates a laser ray, which is swept over the specimen thanks to scanning galvanometer mirrors [87]. The specimen is optically sectioned with laser rays, which create many series of sections. Those created sections will be turned into two—dimensional images later on, that are eventually going to be reconstructed into a three-dimensional reconstruction (3D convolution) of our sample. [88] Those reconstructions provide us with high quality information about the biofilm structure and allow us to visualize the dynamics of the bacterial populations forming biofilm, because the fractions of live/dead cells in the biofilm can also be determined (which is not possible with SEM) [89].

Confocal laser scanning microscopy is a viable tool that is not particularly involved in the process of diagnosing biofilm-related infections. It can be very useful in studying the dynamics of biofilm-forming bacteria on materials used in orthopedics [90]. Understanding the composition and architecture of biofilms is essential in fighting associated infections. A study from 2016 proposed and analyzed whether confocal laser scanning microscopy can be used to highlight and measure the biofilm-forming process on implant materials. This particular study used sandblasted titanium discs, with *Staphylococcus aureus*, *Pseudomonas aeruginosa* and *Candida albicans* biofilms attached. The results showed that CLSM was effective in analyzing biofilms on implant materials. Although the structure of biofilm makes the analysis difficult to perform and only small samples can be visualized, it provides satisfactory results by quantifying all the cells in the biofilm. [91]. This form of biofilm analysis can be used when analyzing biofilm in PJIs on removed implants. Moreover, it can be applied in studies of new ways (compounds) to eradicate biofilm-forming bacteria.

### 7.3. Methylene Blue (MB) as a Disclosing Agent in Biofilm Related Infections Treatment

Methylene blue (MB) is a phenothiazine dye that has been widely used in many areas of medicine for many years, particularly for staining cells. It has many other applications, such as the treatment of methaemoglobinaemia, and also serves as a bioactive photosensitizer in modern photodynamic therapy, used for example in oncology [92,93]. Those characteristics of this substance can be also exploited in orthopedic biofilm related infections treatment.

Most PJIs require a revision surgery in order to treat them. Those surgeries include the removal of pus, necrotic tissue, dead bone, abscess membranes and granulation tissue. In terms of endoprostheses, they usually demand extraction [94]. In many of those surgeries, the decision about which tissues should be removed relies on the experience of the orthopedic surgeon, and that is where methylene blue has potential. In a study from 2019 performed by Shaw et al., it was put to the test whether MB can potentially serve as an intraoperative biofilm visualizing technique to eventually help the surgeon determine which tissues should be extracted. The study took place on *Staphylococcus aureus* and *Pseudomonas aeruginosa* biofilms. The biofilms were grown for 90 days on titanium, cobalt chromium, polyether ether ketone (PEEK), or polyethylene coupons. Those materials are commonly used in orthopedic implants [95]. Later on, the biofilm samples were stained with MB solutions (0.005% and 0.01%). Tissues taken from adult sheep were also stained to determine if the MB substance does tarnish tissues. To see the amount of methylene blue bound to those biofilms, transmission photometry was applied. The results showed that MB can successfully serve as an disclosing agent for the above-mentioned biofilms. What is more, the MB staining did not affect the materials used for biofilm colonies and it also did not substantially stain healthy tissues outside the meniscus and articular cartilage, which probably limits the use of this method in the future to cases involving total joint arthroplasty and joint fusion (where the meniscus and cartilage are not present). This study showed that methylene blue can serve as an effective disclosing agent in vitro when it comes to *Pseudomonas aeruginosa* and *Staphylococcus aureus* biofilms, but more in vivo studies are needed before its further implementation [96]. Another study involving sixteen total knee arthroplasty patients with PJIs tried applying MB during the treatment. During surgical procedures, dilute methylene blue (0.1%) was introduced into the knee. Subsequently samples of stained and not-stained tissues were analyzed with PCR. The results showed that more bacteria were found in the samples previously stained with MB [97]. Although methylene blue, throughout the years, has been used in many different ways in biology, chemistry and medicine, there are studies mentioning its potentially harmful effect on humans. Its toxic effects depend on the applied dose. Toxic doses of MB can cause some serious health issues, such as hemolysis, paradoxical methemoglobinemia, nausea and vomitus, chest pain, dyspnea, and hypertension [98,99]. This shows that its potential application in visualizing biofilm in PJIs would mainly depend on its dosage. Nevertheless, to prove its potential or exclude its application, more in vivo studies must take place. 

See the comparison of all methods in Table 1 and Table 2.

## 8. Antibiotic Therapy

Antibiotics play a partial role in managing prosthetic joint infections (PJI) and are intricately connected to the employed surgical approach. The effectiveness of antibiotics is significantly reliant on the selection and quality of the surgical procedure, as well as the precision of the microbiological diagnosis [100]. Diagnosing prosthetic joint infections in a clinical setting can be difficult due to the prolonged and subtle progression of chronic infections, challenges in culturing low-virulence bacteria, and the diverse range of microorganisms and antibiotic-resistant phenotypes that may be implicated [101,102]. The accurate identification of the involved microorganism(s) and their antibiotic susceptibilities is vital for the successful treatment of these patients [103].

An antibiotic labeled as “anti-biofilm active” is traditionally defined as an antibiotic that effectively permeates the biofilm, demonstrating its ability to eliminate bacteria embedded within it. Furthermore, in the context of treating prosthetic joint infections, it is crucial that the antibiotic efficiently penetrates the bone and, in the case of oral antibiotics, possesses sufficient bioavailability [104]. For instance, β-lactam antibiotics, particularly in their oral forms, are typically discouraged as the primary treatment option. This is due to the limited oral bioavailability of β-lactams, and their generally bactericidal nature. Consequently, their effectiveness in chronic biofilms is hindered by bacteria that are not actively dividing [105]. The emergence of antimicrobial tolerance during the maturation of the biofilm adds complexity to the treatment of prosthetic joint infections (PJIs). Effectively managing PJIs requires appropriate source control, antimicrobial therapy utilizing “anti-biofilm active” antibiotics, and extended durations of antibiotic treatment. 

### 8.1. Types of Therapy

Effective treatment demands comprehensive antibiotic coverage, particularly in the initial phase, specifically during the first few weeks following surgery. 

The intravenous route of administration enables the application of elevated doses, circumvents the hindrance to bioavailability posed by the intestinal and hepatic first-pass effect for specific drugs [106], and mitigates issues related to gastrointestinal intolerance and malabsorption. This is particularly crucial in the context of patients around 60–80 years old, who often receive concomitant medications, including heavy metals like iron, known to reduce drug absorption [107]. The current inclination is to reduce the duration of IV therapy in order to minimize the complications associated with catheters, shorten hospitalization, and reduce associated expenses [108]. The Oral Versus IntraVenous-Antibiotic (OVIVA) trial, a randomized controlled study conducted in the United Kingdom, compared patients undergoing treatment for non-severe Bone and Joint Infections (BJIs) with either 6 weeks or 1 week of intravenous (IV) antibiotics, followed by oral administration. The participants included individuals with various types of BJIs, such as osteomyelitis, arthritis, Prosthetic Joint Infections (PJI), and others, both with and without the use foreign bodies. The treatment approach involved either retaining or removing implants. After a one-year follow-up, there was no significant difference in treatment failures between the two groups [109]. The careful consideration of various factors is essential to determine the suitability for transitioning to oral treatment. The bacteria involved must be susceptible to oral antibiotics that are not only well-tolerated but also readily absorbed by the intestine. Additionally, it is crucial that the drugs are either not metabolized or only poorly metabolized in the intestine and liver to guarantee therapeutic concentrations in both plasma and tissues. Patients confronting challenging or antibiotic-resistant bacteria, undergoing complex surgical procedures such as extensive bone reconstruction, or dealing with issues like gastrointestinal intolerance or malabsorption may require an extended or continuous administration of intravenous (IV) antibiotics throughout the entire course of treatment. [110]. 

### 8.2. Bone and Joint Penetration

The effective penetration of antibiotics into the bone and joint tissue is a significant factor in the treatment of Prosthetic Joint Infections (PJI), particularly in the context of chronic infections with bone engagement. Antibiotics with good bone tissue penetration profiles comprise amoxicillin, piperacillin/tazobactam, flucloxacillin, cloxacillin, cephalosporins across all four generations, carbapenems, aztreonam, aminoglycosides, fluoroquinolones, doxycycline, vancomycin, linezolid, daptomycin, clindamycin, trimethoprim/sulfamethoxazole, fosfomycin, rifampin, dalbavancin, and oritavancin. Exceptions to this trend are observed with penicillin and metronidazole, both demonstrating suboptimal penetration into bones. [111]. The typical range for antibiotic bone/plasma ratios is 0.1 to 0.3. Macrolides, rifampicin and fluoroquinolones show detectably better results [112,113]. 

### 8.3. Treatment Duration

The standard duration for treating Prosthetic Joint Infections (PJI) has been a minimum of three months. A multicenter retrospective study, which included 87 hip or knee PIJ episodes in patients undergoing DAIR (debridement, antibiotics and implant retention) from three university hospitals in France and Switzerland showed not significantly different results for those receiving 6 versus 12 weeks of antibiotic therapy [114]. In a Spanish study involving 63 patients with acute staphylococcal Prosthetic Joint Infections (PJIs) treated with DAIR, including a combination of levofloxacin and rifampicin, findings indicated that an 8-week antibiotic regimen was comparable in efficacy to the conventional 3–6 months of treatment. [115]. The randomized controlled trial performed with 123 patients in Geneva University Hospital showed that the recurrence rates did not differ significantly between 4 and 6 weeks of systemic targeted antibiotic treatment following the removal of the infected implant [116]. In summarizing the referenced research, it is evident that antibiotic therapy must be tailored to align with the chosen surgical strategy. Taken together, the recommended duration for antibiotic treatment in cases of acute Prosthetic Joint Infections (PJIs) treated with debridement, antibiotics, and implant retention (DAIR) is generally 3 months, although it can be shortened for cases with a lower risk of failure. In situations involving revision surgery, a 6-week course of antibiotic treatment is likely adequate, except there are additional factors such as extensive bone graft, extensive osteomyelitis or immunosuppression, and in these cases 12-weeks antibiotic therapy is more effective [117,118].

### 8.4. Antibiofilm Activity

A traditional understanding of an “anti-biofilm active” antibiotic is one that effectively infiltrates the biofilm and showcases its ability to eliminate the bacteria entrenched within it. Furthermore, in managing prosthetic joint infections, it is essential that the antibiotic effectively permeates the bone tissue. As mentioned previously, penicillin and metronidazole demonstrated suboptimal penetration into bone tissue. 

Antibiotics might encounter challenges diffusing through the matrix of the biofilm and achieving effective concentrations, particularly within the inner core. Crucial for biofilm survival are the mechanisms including protection against oxidative stress specific to biofilms, the exclusive expression of efflux pumps within biofilms, and the shielding provided by matrix polysaccharides [119]. The exopolysaccharide matrix controls diffusion and serves as a primary barrier, inhibiting the entry of polar and charged antibiotics. The structure of biofilms enables efficient horizontal gene transfer among bacteria, a process that significantly contributes to the development of antibiotic resistance. The effectiveness of antibiotics within a biofilm can be measured using parameters like the minimal biofilm inhibitory concentration (MBIC) and minimal biofilm eradication concentration (MBEC) [120]. The penetration of antibiotics used in PJI through the biofilm extracellular matrix and penetration into the bone and joint has been summarized in the Table 3.

Max Geller et al., in results from a prospective cohort study regarding a comparison between treatment with biofilm-active antibiotics and conventional treatment without them, concluded that an anti-biofilm active antibiotic led to notably enhanced treatment outcomes for knee prosthetic joint infections, resulting in improved infection resolution, reduced pain levels, and better joint function [121].

In laboratory models assessing the effectiveness of single-drug therapy, it was observed that eliminating bacteria embedded within biofilms is particularly challenging, especially in mature biofilm formations. The bacteria within aging biofilms have been observed to have a lower susceptibility to antimicrobial agents compared to those in younger biofilms [4]. The use of combined therapy seems to be the right solution for biofilm eradication. Parra-Ruiz et al., on an in vitro model of *Staphylococcus aureus* biofilm, investigated that the activity of linezolid and high-dose daptomycin alone is much lower than in combination [122]. A similar in vitro study by Aktas et al., which compared the activity of daptomycin combined with dalbavacin and linezolid, and dalbavicin with linezolid against MRSA strains, showed that the rates of synergistic effects were 67% for daptomycin combined with dalbavancin and linezolid, and 60% for dalbavancin combined with linezolid [123].

### 8.5. Empiric Therapy

Empirical antibiotic treatment, commonly involving a broad-spectrum β-lactam with strong efficacy against Gram-negative rods (such as piperacillin–tazobactam or cefepime) and a drug active against Gram-positive cocci, including methicillin-resistant staphylococci (like daptomycin or vancomycin), is frequently administered at the beginning of treatment while awaiting the outcomes of sample cultures. Unfortunately, using such a therapy routinely contributes to the acquisition of resistance to commonly used antibiotics [124]. American guidelines include piperacillin-tazobactam or ceftriaxone and vancomycin as a standard empiric therapy [125,126]. A retrospective cohort study showed that the most extensive antimicrobial coverage for all typical pathogens in early Prosthetic Joint Infections (PJI) was achieved by combining vancomycin with ciprofloxacin or a third-generation cephalosporin [127]. The recommendations given by Izakovicov et al., based on the analysis of the current concepts of treatment and diagnosis of PJI, assume the use of ampicillin/sulbactam or amoxicillin/clavulanic acid IV with additional vancomycin in septic patients, known MRSA carriers, those with multiple previous surgeries, and suspected low-grade infection as a recommended empirical treatment. This review also highlights the importance of incorporating a biofilm-active antibiotic into the regimen to achieve the best possible outcomes [128]. The Italian guideline from 2009 suggested a parenteral therapy composed of amoxicillin or ceftriaxone +/− rifampicin as an empirical therapy for patients without a risk factor for MRSA and vancomycin or teicoplanin or linezolid, or daptomycin +/− rifampicin for cases with risk factors for MRSA [129]. Bassetti et al. decided to update these guidelines based on their own experiences. The Udine strategy algorithm for the management of prosthetic joints suggests the use of a combination therapy of glycopeptide or lipopeptide + rifampicin with quinolone (levofloxacin) if there is a risk of Gram-negative bacteria, used while waiting for culture data [130]. The French empirical treatment strategy is based on the use of vancomycin in combination with a broad-spectrum beta-lactam (mainly piperacillin-tazobactam). However, vancomycin and tazobactam used together enhance the nephrotoxic effect, causing a high rate of AE, and especially AKI. This should be taken into account, especially in patients with other factors. A study of the tolerability of empirical treatment in PJI from Lyon BJI Study Group suggested vancomycin-cefepime (with or without metronidazole) as a more renal-sparing treatment [131]. Research carried out based on the medical records of 75 patients in Centro Hospitalar do Porto—Hospital Santo António also showed proposed empirical evidence based on their experiences. They suggested vancomycin for acute infection, vancomycin (Gram-positive active) and carbapenem (Gram-negative active) for chronic and hematogenous types. Treatment should be immediately modified after determining the group of bacteria—Gram-negative, Gram-positive, or mixed [132]. An analysis of the PIANO cohort enabled the following recommendations to be made. For early post-operative treatment, the use of vancomycin plus a Gram-negative agent (ciprofloxacin, cefepime or gentamicin, according to local antimicrobial guidelines) is recommended. In the case of late acute PJI, cefazolin is also recommended. If a chronic non-septic form occurs, antibiotic therapy should be suspended until the culture results are obtained [133]. A study conducted at the Orthopedic Surgery Center in Zhengzhou, China from January 2012 to December 2018 showed that the empirical antibiotic therapy should include a combination of vancomycin with either cephalosporin, levofloxacin, or clindamycin. Once the pathogen is identified, the treatment should be individualized immediately [134]. A systematic review and meta-analysis from 2024 by Ya-Hao Lai et al. confirmed the use of vancomycin with other locally selected antibiotics, such as cephalosporins, as an effective way to control PJI [135]. The Korean clinical guidelines suggest vancomycin or teicoplanin optionally with the addition of ceftazidime or cefepime as the preferred therapy. Antimicrobial treatment should be initiated empirically only after obtaining cultures from blood, abscesses, and bone tissue samples [136]. An Indian review emphasized the need to introduce empirical therapy preventively during all orthopedic surgeries. They recommend introducing therapy 15–30 min before the start of each procedure and continuing it during the operation. Routine treatment includes cefuroxime 1.5 g IV Stat and BD for 24 h or cefazolin 2 g IV Stat and BD for 24 h. The therapies of choice in implant surgery infections are ceftriaxone IV and vancomycin IV for 4 weeks [137]. A South African retrospective cross-sectional study, based on data collected from the Charlotte Maxeke Johannesburg Academic Hospital hip and knee and the Johannesburg Orthopedic hip and knee databanks, led to the establishment of the recommended therapy, which should consist of meropenem or gentamicin, vancomycin and rifampicin. According to the authors, such a recommendation is to ensure coverage of the widest possible spectrum, and thus to ensure the maximum possible effectiveness of eradication of the infection. They also emphasize the difference between the recommended treatment in PJI and the American or European guidelines [138]. Pecora JR et al., from Sao Paulo, Brazil, indicated that to preserve the joint prosthesis, a broad-spectrum antibiotic should be used based on hospital microbial profiles, and surgical debridement should be used to identify the causative factor and initiate targeted therapy immediately. It is worth noting that they did not propose any specific treatment regimen recommended to all centers, but determined the treatment based on local microbiological profiles as the most appropriate [139]. In summary, empiric treatment should be adapted to local experience and modified as quickly as possible to provide the optimal treatment for a given case. It is often standard practice to initiate treatment to avoid the development of infection. The selected procedures of treatment vary slightly between centers, as shown in the Figure 5.

### 8.6. Targeted Therapy

As mentioned above, the inclusion of targeted therapy should be the gold standard of treatment. An antibiogram is key to selecting the appropriate drug and the route and time of its administration. Numerous studies have shown which treatment regimens are usually the most effective for infections with specific bacteria. The most frequently used targeted treatment is presented in Table 4.

#### 8.6.1. *Staphylococuus* spp.

Staphylococci are overwhelmingly the predominant microorganisms found in PJIs, with *S. aureus* and *S. epidermidis* being the prevailing species detected most frequently [103]. The preferred treatment for methicillin-susceptible staphylococcal PJIs involves the initial administration of intravenous anti-staphylococcal β-lactam antibiotics, such as oxacillin or cefazolin [99]. A systemic review and meta-analysis showed that the addition of rifampicin increased the therapy success rate by 10% [140].

#### 8.6.2. *Streptococcus* spp.

Streptococci are the next most frequently encountered microorganisms, comprising approximately 9–16% of cases, and they have the capability to produce biofilms as well, but this process is comparatively less extensive than that of *S. aureus* [141]. International guidelines suggest starting intravenous therapy with high-dose penicillin or ceftriaxone. Additionally, amoxicillin is frequently administered and is the preferred initial treatment according to French guidelines. It is also the primary choice for oral therapy switching, typically prescribed at a dosage of 2–3 g three times a day [142]. The retrospective study revealed that the addition of rifampicin did not provide any advantage in patients who were already receiving a β-lactam [143].

#### 8.6.3. *Enteroccous* spp.

Enterococci are responsible for 2 to 11% of all prosthetic joint infections (PJI) [144]. International guidelines suggest using high doses of intravenous penicillin or ampicillin. The preferred choice for prescribers often includes combination therapy with gentamicin or rifampicin [102]. 

After 2 to 4 weeks of intravenous therapy, a transition to oral amoxicillin administration is possible. For patients allergic to penicillin, intravenous high-dose vancomycin or daptomycin are alternatives. The only oral alternative is linezolid, yet its toxicity profile may constrain its long-term usage [145].

#### 8.6.4. *Pseudomonas aeruginosa*

Preventing and treating infections caused by *Pseudomonas* spp. poses significant challenges, mainly due to their prevalent presence in hospital environments and their high likelihood of developing drug resistance. Effective agents against *Pseudomonas aeruginosa* comprise ceftazidime, cefepime, piperacillin–tazobactam, and carbapenem. International guidelines prioritize cefepime or meropenem as the preferred choices, while ciprofloxacin and ceftazidime are considered alternative options [109]. It appears that an effective initial intravenous antibiotic treatment lasting at least 3 weeks is necessary, succeeded by a course of oral ciprofloxacin spanning a total duration of 3 months [146].

#### 8.6.5. *Enterobacteriacea*

*Enterobacteriaceae* can cause infections independently, as well as contribute to mixed infections. International guidelines for treating *Enterobacteriaceae* PJIs recommend IV β-lactam, selected according to in vitro susceptibility, or oral ciprofloxacin 750 mg twice a day [102]. The large retrospective observational study in 16 Spanish hospitals showed that the use of ciprofloxacin resulted in a clinical success rate of 79% [147]. Furthermore, the study performed at Nantes University Hospital in France led to the conclusions that intravenous β-lactam-based therapy should be regarded as an alternative for patients who are unable to receive fluoroquinolones [148].

#### 8.6.6. *Cutibacterium* spp.

*Cutibacterium* spp. are susceptible to many antibiotics routinely and widely used. These infections are commonly treated with intravenous β-lactam antibiotics and clindamycin, followed by oral clindamycin or amoxicillin [149]. The guidelines in the United States advise penicillin or ceftriaxone as the primary treatment, and vancomycin or clindamycin are suggested as a secondary treatment path [99]. According to the French guidelines, the recommended initial treatments should include amoxicillin, cefazolin, or clindamycin [150]. *C. acnes* is predominantly found in infections of shoulder and hip implants, while *C. avidum* is commonly linked with infections in hip arthroplasty. A noteworthy multicenter retrospective study, which analyzed data on the treatment course of 187 patients with *Cutibacterium* PJI, showed that the inclusion of rifampicin in the antibiotic treatment correlated with improved clinical results only when the implants were either removed or replaced, and patients received antibiotic therapy for a minimum of six weeks. It was emphasized that it is not recommended as a routine therapy [151].

#### 8.6.7. Other Anaerobes

*Bacteroides* spp., *Clostridium* spp. are more often a factor of mixed infection, but sometimes they also occur as a single causative factor. The medical records of 17 episodes of *Bacteroides* spp. total hip or knee arthroplasty infection seen at the Mayo Clinic in Rochester between 1969 and 2012 analyzed in the retrospective cohort study from 2017 indicated that most of the isolates showed susceptibility to metronidazole, imipenem, tetracyclines, and clindamycin. Eight patients received oral therapy following the initial intravenous course [152]. In clostridial PJIs, the use of penicillin, clindamycin, and metronidazole is the most common procedure, supported by the general susceptibility profile [153].

**Table 4 microorganisms-12-01198-t004:** The most frequently used treatment and the possible route of targeted administration, depending on the microorganism.

Bacteria	Initial IV Therapy	Possible Oral Switch	References
Methicillin-susceptible *Staphylococcus*	Oxacillin+ rifampicin Cefazolin+ rifampicin Levofloxacin + rifampicin	Levofloxacin + rifampicin Ciprofloxacin + rifampicin	[99,109]
Methicillin-resistant *Staphylococcus*	Vancomycin (most recommended) + rifampicin or minocycline Daptomycin or linezolid+ rifampicin or minocycline	No oral switch	[103,140,149]
*Streptococcus* spp.	Penicillin Ceftriaxone Amoxicillin	Amoxicillin	[141,142,143]
*Enterococcus* spp.	High-dose penicillin or ampicillin +gentamicin or rifampicin Amoxicillin +/− initial gentamicin	Amoxicillin (worse bioavailability) Clindamycin Linezolid	[99,144,145]
*Pseudomonas aeruginosa*	Cefepime Meropenem Ceftazidime Ciprofloxacin	Ciprofloxacin	[109,146]
*Enterobacteriaceae*	Cefotaxime Ceftriaxone	Ciprofloxacin Levofloxacin	[99,147,148]
	Ciprofloxacin		[153]
*Cutibacterium* spp.	Amoxicillin Clindamycin Cefazolin	Amoxicillin Clindamycin	[99,150,152]
	Penicillin Ceftriaxon		[151]
*Bacteroides* spp.	Metronidazole Imipenem Clindamycin	Metronidazole Imipenem Clindamycin	[150,152]
*Clostridium* spp.	Penicillin Clindamycin Metronidazole	Clindamycin Metronidazole	[153]

## 9. Conclusions

Our review summarizes the current knowledge of microbiological methods for diagnosing biofilm found especially in PJIs. We have included both widely used methods and modern ones, whose use on a larger scale in healthcare systems around the world could be of interest to the medical staff in the near future to improve diagnostic and cure standards. 

In our work, we also noted that science is currently slowly moving away from the traditional, static research model of biofilm formation towards microfluidic conditions, which better reflect the dynamics of our body. It turns out that these are completely different models and can often explain the rapid development of infection, including sepsis. We introduced the idea of using a flow model as a potential novelty in demonstrating the presence of biofilm and dynamics of infection. The characteristics of the effectiveness of other available methods are also included, which was intended to make our review particularly useful to practicing doctors.

We have collected information about the available materials currently or potentially used in orthopedy that prevent the occurrence of biofilm.

The current knowledge about the requirements that a chemotherapeutic agent must meet to be useful in combating postoperative infections in orthopedics was presented. We took into account the practical guidelines regarding currently conducted therapeutic regimens, both empirical therapy and the most common antibiotics used in the targeted therapy of specific biofilm-forming bacteria.

This review aims to collect potential microbiological diagnostic paths and the selection of antibiotic therapy in order to present, in a clear form, the current knowledge in the field of infections with biofilm-forming bacteria in orthopedy for specialists diagnosing and treating these types of infections, especially PJIs.

## Figures and Tables

**Figure 1 microorganisms-12-01198-f001:**
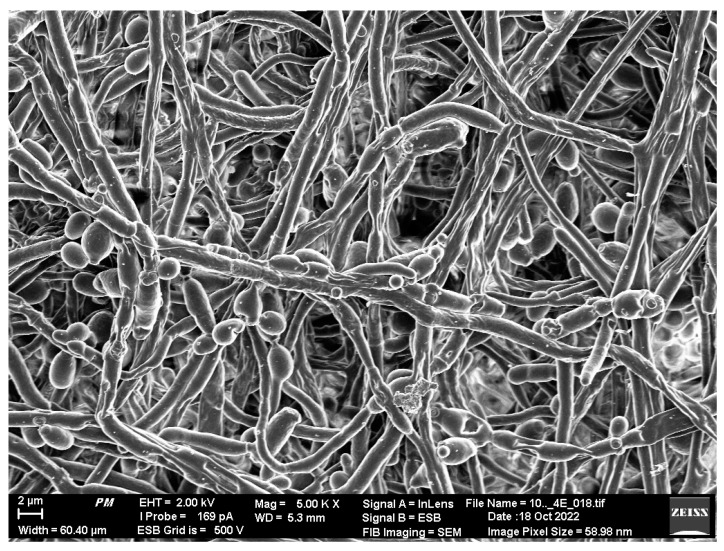
The mature biofilm of *Candida albicans* on metal surface analyzed using Scanning Electron Microscope Auriga 60 (SEM). Scale bar = 2 µm.

**Figure 2 microorganisms-12-01198-f002:**
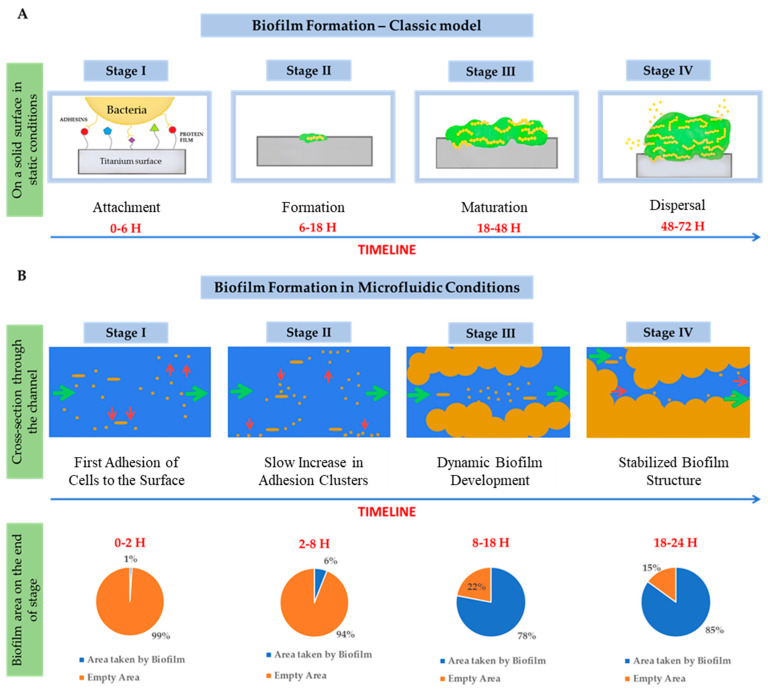
The classic, static model of biofilm formation (**A**—top) and the model of biofilm formation in microfluidic conditions (**B**—bottom).

**Figure 3 microorganisms-12-01198-f003:**
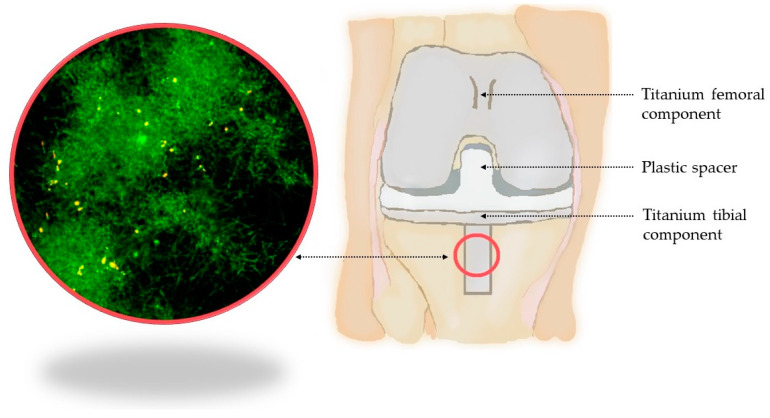
Basic structure of the knee prosthesis and biofilm structure of *C. albicans* on titanium implant.

**Figure 4 microorganisms-12-01198-f004:**
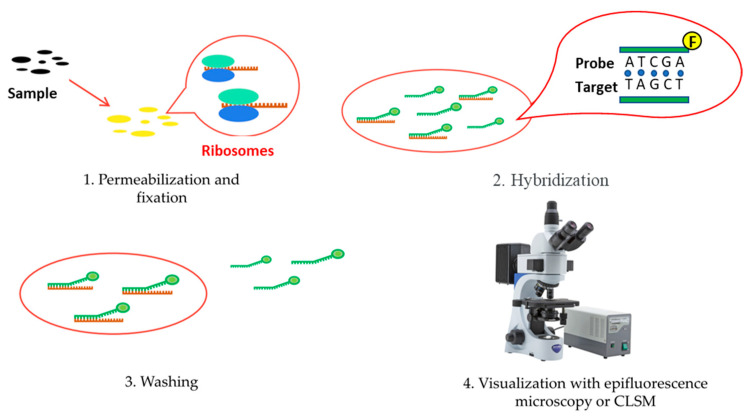
Fluorescence in situ hybridization (FISH) procedure: 1. Permeabilization and fixation using chemical fixatives. 2. Hybridization with the target marker. 3.Washing to reduce background fluorescence. 4. Visualization with epifluorescence microscopy or confocal laser microscopy.

**Figure 5 microorganisms-12-01198-f005:**
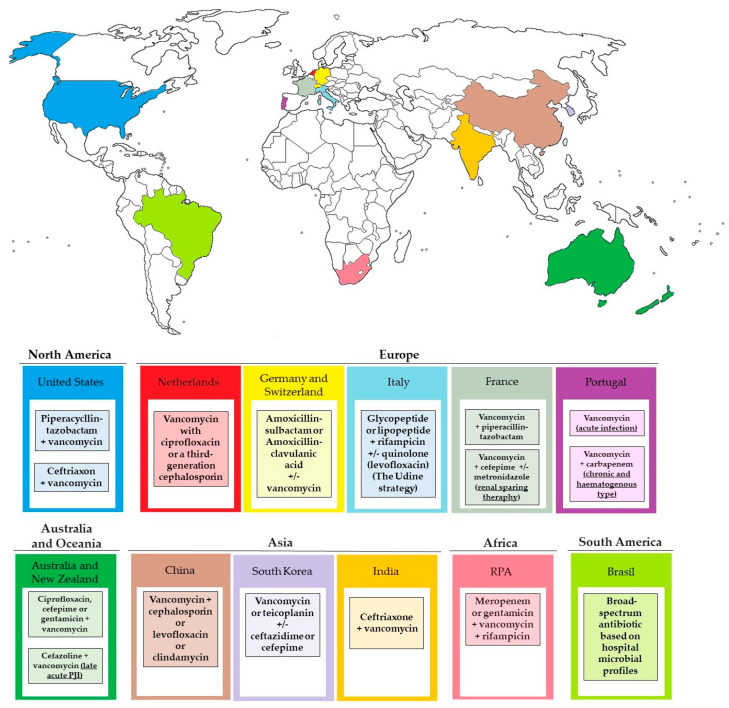
Selected practical recommendations regarding the proposed effective empirical treatment according to data included in the standards of practice of various centers around the world (The US, The Netherlands, Australia and New Zealand, India, Portugal, South Korea).

**Table 1 microorganisms-12-01198-t001:** Comparison of microbiological methods used in the detection of biofilm-forming microorganisms with particular advantages and disadvantages.

Diagnostic Method	Advantages	Disadvantages	References
Direct microbiological culturing	- Low cost - Common availability - Does not require any specific staff education or equipment	- Risk of results being false-negative - Antibiotic therapy before obtaining samples may lead to false-negative results - Risk of contamination, which in some cases leads to misdiagnosis - Method takes time	[45,46,47,48,49,50,51]
Culturing sonication fluid	- Higher sensitivity and specificity when compared to standard culture methods - Inoculating sonication fluid into blood culture bottles can enhance its diagnostic value - Good results when dealing with patients that were under antimicrobial therapy	- Requires specialized sonication equipment - High risk of contamination during the whole process	[52,53,54,55,56,57,58]
DTT pre-treatment method	- Low cost - High specificity and sensitivity (superior to standard culturing and similar to sonication) - Less chance of contamination compared with sonication method	- Requires more study - While generally safe, DTT can be toxic if ingested or absorbed through the skin in large amounts. Hence, proper safety precautions are needed	[59,60,61]
BioTimer Assay (BTA) method	- Low cost - Can be considered as a valuable tool for analyzing antibiotic susceptibility	- Incapable of identifying microbial species	[62,63]
Agar encasement culturing method (AECM) with Candle dip method	- Its combination overcomes many limits of AECM - It allows for biofilm growth without any disruption - It is possible to grow species that are sensitive to oxygen - Provides viable information about the locations on endoprostheses prone to biofilm formation and how they change with the time of incubation	- Specific conditions, agar and plenty of space are required for agar rinsing and incubation of whole explanted material - Time-consuming - Multiple agar coating of the material is required so that it does not dry out.	[64]
Specific PCR	- Readily available in many laboratories and possesses high sensitivity in determining PJI-causing organisms - Multiplex RT-qPCR designed to quickly detect *Cutibacterium* infections (most common in shoulder infections)	- Specific PCR only targets solitary bacterial species or closely related bacterial groups - Potential false-positive results while applying RT-qPCR for *Cutibacterium*	[65,66,67,68,69]
Broad-range PCR	- Similar to specific PCR but it targets more microbial species so it is more useful in periprosthetic joint infections, which can be caused by various microorganisms - Laboratories may create their own combinations for broad-range PCR based on the most common PJI causes in their hospitals - There are commercial kits specifically for detecting PJI-causing species, like Unyvero i60^®^ multiplex PCR	- It can only identify targeted species, sometimes potentially excluding the cause of PJIs and providing negative results - Specialized personnel, prepared laboratories and sufficient sample collection is needed - Risk of contamination	[70,71,72,73]
Next generation sequencing	- Higher sensitivity and specificity than PCR and culture methods - Simultaneously sequences millions to billions of fragments in one reaction system	- Requires more studies in terms of its application in PJIS diagnosing - Lack of standardization in terms of thresholds for single bacterial species percentage - Some differences between studies regarding this methods sensitivity and specificity - Risk of contamination, resulting in false-positive results	[74,75,76]
Fluorescence in situ hybridization (FISH)	- Provides visualization, quantification and identifies various targets involved in biofilm formation - Can be combined with CLSM to visualize fluorescently labeled samples	- Requires specific preparation of the sample (permeabilization and fixation, hybridization, washing) for its visualization to be successful - Only fluorescently tagged areas are analyzed	[77,78,79,80]

**Table 2 microorganisms-12-01198-t002:** Comparison of methods useful in visualizing biofilm-forming microorganisms with particular advantages and disadvantages.

Visualization Method	Advantages	Disadvantages	References
Scanning electron microscopy (SEM)	- Powerful imagining technique which provides detailed, three-dimensional images of a sample’s surface, resolving even singular bacterial cells	- Only small samples can be visualized - Requires specific equipment which is not widely available in most laboratories - Samples, before visualizing, need to be chemically fixated and dehydrated	[81,82,83,84]
Confocal laser scanning microscopy (CLSM)	- Provides high quality visualization of the dynamics of biofilm formation on orthopedic implants - Fractions of live/dead cells in biofilm can be determined which is not possible with SEM - Applying specific software allows for the quantification of all cells in samples of biofilm formed on implants	- Only small samples can be visualized - Requires specific equipment which is not widely available in most laboratories	[85,86,87,88,89,90,91]
Methylene blue (MB)	- Potential in intraoperative application as a disclosing agent for areas that should be surgically removed - In vitro studies proved its successful staining of *Staphylococcus aureus* and *Pseudomonas aeruginosa* biofilms - In in vitro studies, MB did not substantially stain healthy tissues	- Lack of in vivo studies to determine any undesirable effects of its application in PJI treatment - Potential toxic effects on human body	[92,93,94,95,96,97,98,99]

**Table 3 microorganisms-12-01198-t003:** Penetration of antibiotics commonly used in PJI through the biofilm extracellular matrix of *S. aures*, *S. epidermidis*, *P. aeruginosa*, *E.coli* combined with penetration into the bone and joint.

Bacteria	Antibiotic	Penetration through the Biofilm Extracellular Matrix	Penetration into the Bone and Joint
*S. aures*	Amikacin	+	+
	Cefotaxime	Reduced	
	Ciprofloaxacin	+	
	Oxacillin	Reduced	
	Vancomycin	+/Reduced	
	Linezolid	+	
*S. epidermidis*	Amikacin	+	
	Cefotiam	+	
	Cefotaxime	Reduced	
	Ciprofloaxacin	+	
	Daptomycin	+	
	Linezolid	+	
	Ofloxacin	+	
	Oxacillin	Reduced	
	Rifampicin	+	
	Vancomycin	Reduced	
*P. aeruginosa*	Amikacin	Reduced	
	Amoxicillin- clavulanic acid	+	
	Ciprofloaxacin	+	
	Gentamicin	Reduced	
	Imipenem	+	
	Levofloxacin	+	
	Piperacillin	+/Reduced	
	Fosfomycin	+	
*E.coli*	Amoxicillin- clavulanic acid	+	
	Ciprofloaxacin	+	
	Fosfomycin	+	
		References:	[110,120,121]

## Data Availability

Not applicable.

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
