# Peer review of "Modern Microbiological Methods to Detect Biofilm Formation in Orthopedy and Suggestions for Antibiotic Therapy, with Particular Emphasis on Prosthetic Joint Infection (PJI)"

_microorganisms, 2024, doi:10.3390/microorganisms12061198_

Round 1

Reviewer 1 Report

Comments and Suggestions for Authors

Dear Authors!

Thank you very much for your review scientific article, which touches on important issues in bone orthopedics. Improving the quality of the implant and the degree of biocompatibility with the patient's body is critical for successful recovery.

1) Could the Authors add a short note about ion-doped hydroxyapatite (or others calcium phosphate based compounds), which, due to impurity centers, can have pronounced antimicrobial properties. Bone implants based on calcium phosphates are good because they have a phase and elemental composition close to human bone tissue. Thus, they can serve as endogenous prostheses with excellent adhesive (surface) properties, stimulating the growth of osteoblasts upon contact with the damaged area of ​​the patient's hard tissue.

2) How does surface morphology (roughness, size/number of pores, etc. studied using various microscopies (SEM, TEM)) affect the degree of biofilm formation? It might be worth mentioning this in the article.

There are small typos within the article that need to be corrected (...text... . [refs]. lines 35, 42, 245...  ect. ... 780, 26 times)

Sincerely, Reviewer

Author Response

R1

Dear Authors!

Thank you very much for your review scientific article, which touches on important issues in bone orthopedics. Improving the quality of the implant and the degree of biocompatibility with the patient's body is critical for successful recovery.

A: Thank you very much for such kind opinion.

  • Could the Authors add a short note about ion-doped hydroxyapatite (or others calcium phosphate based compounds), which, due to impurity centers, can have pronounced antimicrobial properties. Bone implants based on calcium phosphates are good because they have a phase and elemental composition close to human bone tissue. Thus, they can serve as endogenous prostheses with excellent adhesive (surface) properties, stimulating the growth of osteoblasts upon contact with the damaged area of the patient's hard tissue.

A: Thank you very much for suggestion. A note about hydroxyapatite has been added.

  • How does surface morphology (roughness, size/number of pores, etc. studied using various microscopies (SEM, TEM)) affect the degree of biofilm formation? It might be worth mentioning this in the article.

A: Thank you very much for suggestion. A information has been mentioned in article.

There are small typos within the article that need to be corrected (...text... . [refs]. lines 35, 42, 245...  ect. ... 780, 26 times)

A: Thank you very much for suggestion. All mistakes has been corrected.

Thank you very much for your positive review and we wish you many scientific successes.

Reviewer 2 Report

Comments and Suggestions for Authors

Authors attempted to perform a review focusing on the modern microbiology methods to diagnose biofilm in orthopedy, with special emphasis on PJI. The topic is not novel, although it can be of interest for practicing clinicians. Nevertheless, in the way it is presented, the manuscript is too long and very difficult to follow and, most important, contains several parts which are out of the main topic of the review (ie, classification, therapy, imaging etc) and render difficult to follow the paper.

Therefore, I strongly suggest to highly revise the paper, focusing only on microbiological diagnosis, which is, in the current form, too short to be complete.

Herein my detailed comments:

-        Abstract should be reordered, indeed as it is, it is quite confusing (first diagnosis, then treatment than pathogens than again diagnosis)

-        “Mentioned above factors show how important it is to detect those complications as soon as possible and how they can be helpful microbiological methods of biofilm detection”: this sentence is not clear

-        If the setting of the review is PJI, please change the title and consider only PJI

-        Furthermore, the main topic of the review relies on modern diagnostic method for diagnosis biofilm infection: therefore, I would suggest to shorten the review, in particular the part of treatment, since it may deserve a review itself. Rather, the problem of biofilm formation and its steps are important points to underline for the main topic of the review; nevertheless, I would change the title of the paragraph 2 as “the problem of biofilm in PJI”.

-        Please check English language

-        “The most common etiological factors associated with biofilms”: it refers to causative microorganisms, please re-write

-        staphylococci coagulase-positive: please correct with the specific name of bacteria

-        Lines 74-76: check English

-        The Candida genus: not correct genus

-        Figure1: is this image property of the authors? Please specify. The same for Figure3.

-        The clinical classification is not needed, authors could shorten the paragraph.

-        Lines 225-226: insert ref

-        Sonication part could be ameliorated a d enlarged, ie by stating the importance of CFU counting and the possibility to inoculate sonication fluid into blood culture to improve sensibility and the possibility to use fluid for additional (ie, molecular) analyses (see at the bottom for relevant ref)

-        There are additional methods, that authors should consider, or at least mention, ie, the DTT  or the Biotimer Assay (see at the bottom for relevant ref)

-        Molecular analysis: authors should comment the specific PCR for C acnes, which is commonly encountered in shoulder infections

-        Rather than only listing the methods, authors should also show these methods applied to the diagnosis of bone/prosthesis infections (ie, studies showing sensibility, eventual comparison with other methods, types of infections, etc)

-        Is SEM useful for the identification/isolation of the causative pathogen for PJIs? If yes, please state in the text, if no please remove it or only mention it in a specific paragraph rather than within other methods able to identify pathogens. The same for CLSM and MB. Authors should also remove them from the Table1 (or create a separate table listing the methods able to visualize biofilm)

-        Lines 429-450: they are useless for the primary aim of the review

-        Table1: cons of sonication can be potential contamination; advantage is to identify polymicrobial infections and possibility to count; no influence of previous antimicrobial therapy. Please revise and be more accurate for all methods

-        Lines 453-590: this paragraph is out of the scope of the review (microbiological diagnosis of PJI) and therefore should be completely removed

-        Lines 591-707: also, this paragraph is out of the scope of the review: it should be dramatically reduced and, if considered necessary, maintained only table2.

-        For the same reasons, authors should eliminate lines 711-862 and the corresponding figure5 and table3: out of the scope of the review

-        Also in the conclusion, authors should focus only on diagnostic methods

-        Authors could include a practical diagnostic algorithm

Overall, reference should be ameliorated. Fundamental missing ref: Trampuz et al. Sonication of removed hip and knee prostheses for diagnosis of infection. N Engl J Med. 2007 Aug 16;357(7):654-63. doi: 10.1056/NEJMoa061588.

Other important, but not limited to, ref that need to be considered:

-        Drago L, Fidanza A, Giannetti A, Ciuffoletti A, Logroscino G, Romanò CL. Bacteria Living in Biofilms in Fluids: Could Chemical Antibiofilm Pretreatment of Culture Represent a Paradigm Shift in Diagnostics? Microorganisms. 2024 Jan 26;12(2):259. doi: 10.3390/microorganisms12020259. PMID: 38399663; PMCID: PMC10892178.

-        Drago L, et al. Use of dithiothreitol to improve the diagnosis of prosthetic joint infections. J Orthop Res. 2013 Nov;31(11):1694-9. doi: 10.1002/jor.22423.

-        Drago L, et al. Dithiotreitol pre-treatment of synovial fluid samples improves microbiological counts in peri-prosthetic joint infection. Int Orthop. 2023 May;47(5):1147-1152. doi: 10.1007/s00264-023-05714-z.

-        Oliva A, et al. Challenges in the Microbiological Diagnosis of Implant-Associated Infections: A Summary of the Current Knowledge. Front Microbiol. 2021 Oct 29;12:750460. doi: 10.3389/fmicb.2021.750460.

-        Di Domenico EG, et al. The Current Knowledge on the Pathogenesis of Tissue and Medical Device-Related Biofilm Infections. Microorganisms. 2022 Jun 21;10(7):1259. doi: 10.3390/microorganisms10071259.

-        Portillo ME, et al. Improved diagnosis of orthopedic implant-associated infection by inoculation of sonication fluid into blood culture bottles. J Clin Microbiol. 2015 May;53(5):1622-7. doi: 10.1128/JCM.03683-14.

-        Watanabe S, et al. Differences in Diagnostic Sensitivity of Cultures Between Sample Types in Periprosthetic Joint Infections: A Systematic Review and Meta-Analysis. J Arthroplasty. 2024 Mar 13:S0883-5403(24)00232-8. doi: 10.1016/j.arth.2024.03.016. Epub ahead of print. PMID: 38490568.

-        Li C, et al. Meta-analysis of sonicate fluid in blood culture bottles for diagnosing periprosthetic joint infection. J Bone Jt Infect. 2018 Dec 24;3(5):273-279. doi: 10.7150/jbji.29731.

Comments on the Quality of English Language

See the detailed comments

Author Response

R2

Authors attempted to perform a review focusing on the modern microbiology methods to diagnose biofilm in orthopedy, with special emphasis on PJI. The topic is not novel, although it can be of interest for practicing clinicians. Nevertheless, in the way it is presented, the manuscript is too long and very difficult to follow and, most important, contains several parts which are out of the main topic of the review (ie, classification, therapy, imaging etc) and render difficult to follow the paper.

A: Thank You very much for review and your suggestions. The article is long because the topic is very broad. We shortened the work as recommended. We decided to focus on microbiological diagnostics in the context of biofilm and leave the important aspect of therapy with antibiotics, which answers the questions of many clinicians in medical practice.

Therefore, I strongly suggest to highly revise the paper, focusing only on microbiological diagnosis, which is, in the current form, too short to be complete.

A: Thank You very much for your suggestions. It has been changed.

Herein my detailed comments:

-Abstract should be reordered, indeed as it is, it is quite confusing (first diagnosis, then treatment than pathogens than again diagnosis)

A: Thank You very much for your suggestions. The abstract has been reorganized.

-Mentioned above factors show how important it is to detect those complications as soon as possible and how they can be helpful microbiological methods of biofilm detection”: this sentence is not clear

A: Thank You very much for your suggestions. The subject has been clarified.

-        If the setting of the review is PJI, please change the title and consider only PJI

A: Thank You very much for your suggestions. The title has been changed.

-        Furthermore, the main topic of the review relies on modern diagnostic method for diagnosis biofilm infection: therefore, I would suggest to shorten the review, in particular the part of treatment, since it may deserve a review itself. Rather, the problem of biofilm formation and its steps are important points to underline for the main topic of the review; nevertheless, I would change the title of the paragraph 2 as “the problem of biofilm in PJI”.

A: Thank You very much for your suggestions. It has been changed. To save a valuable part for clinicians, we added "and antibiotic treatment" in the title.

“Modern Microbiological Methods for Detecting Biofilm Formation and Antibiotic Treatment in Orthopedics Regarding the Aspect of Prosthetic Joint Infections (PJI)”

-        Please check English language

A: Thank You very much for your suggestions. The correction has been done.

-        “The most common etiological factors associated with biofilms”: it refers to causative microorganisms, please re-write

A: Thank You very much for your suggestions. It has been re-written.

-        staphylococci coagulase-positive: please correct with the specific name of bacteria

A: Thank You very much for your suggestions. It has been corrected.

-        Lines 74-76: check English

A: Thank You very much for your suggestions. It has been checked.

-        The Candida genus: not correct genus

A: Thank You very much for your suggestions. It has been corrected.

-        Figure1: is this image property of the authors? Please specify. The same for Figure3.

A: Thank you for this question. All graphics in the work are our own.

-        The clinical classification is not needed, authors could shorten the paragraph.

A: Thank You very much for your suggestions. The fragment has been deleted.

-        Lines 225-226: insert ref

A: Thank You very much for your suggestions. It has been added.

-        Sonication part could be ameliorated a d enlarged, ie by stating the importance of CFU counting and the possibility to inoculate sonication fluid into blood culture to improve sensibility and the possibility to use fluid for additional (ie, molecular) analyses (see at the bottom for relevant ref)

A: Thank You very much for your suggestions. The above-mentioned subject has been expanded.

-        There are additional methods, that authors should consider, or at least mention, ie, the DTT  or the Biotimer Assay (see at the bottom for relevant ref)

A: The previously mentioned methods have been added. Thank You very much for your suggestions.

-        Molecular analysis: authors should comment the specific PCR for C acnes, which is commonly encountered in shoulder infections

A:.  The previously mentioned methods have been added.

-        Rather than only listing the methods, authors should also show these methods applied to the diagnosis of bone/prosthesis infections (ie, studies showing sensibility, eventual comparison with other methods, types of infections, etc)

A: Thank You very much for your suggestions. It has been corrected.

-        Is SEM useful for the identification/isolation of the causative pathogen for PJIs? If yes, please state in the text, if no please remove it or only mention it in a specific paragraph rather than within other methods able to identify pathogens. The same for CLSM and MB. Authors should also remove them from the Table1 (or create a separate table listing the methods able to visualize biofilm)

A: Thank You very much for your suggestions. It has been corrected.

-        Lines 429-450: they are useless for the primary aim of the review

A: Thank You for your suggestions.

-        Table1: cons of sonication can be potential contamination; advantage is to identify polymicrobial infections and possibility to count; no influence of previous antimicrobial therapy. Please revise and be more accurate for all methods

A: Thank You very much for your suggestions. It has been corrected.

-        Lines 453-590: this paragraph is out of the scope of the review (microbiological diagnosis of PJI) and therefore should be completely removed

A: Thank You very much for your suggestions. It has been removed.

-        Lines 591-707: also, this paragraph is out of the scope of the review: it should be dramatically reduced and, if considered necessary, maintained only table2.

A: Thank You very much for your suggestions. It has been corrected.

-        For the same reasons, authors should eliminate lines 711-862 and the corresponding figure5 and table3: out of the scope of the review

A: Thank You very much for your suggestions. We decided to keep the topic of antibiotic treatment at work for practical reasons relevant to clinicians.

-        Also in the conclusion, authors should focus only on diagnostic methods

A: Thank You very much for your suggestions. It has been corrected.

-        Authors could include a practical diagnostic algorithm

A: We will be happy to consider this in the next work related to this topic. Thank You very much for your suggestions.

A: Thank You very much for your suggestions. The references have been added:

-        Drago L, Fidanza A, Giannetti A, Ciuffoletti A, Logroscino G, Romanò CL. Bacteria Living in Biofilms in Fluids: Could Chemical Antibiofilm Pretreatment of Culture Represent a Paradigm Shift in Diagnostics? Microorganisms. 2024 Jan 26;12(2):259. doi: 10.3390/microorganisms12020259. PMID: 38399663; PMCID: PMC10892178.

-        Drago L, et al. Use of dithiothreitol to improve the diagnosis of prosthetic joint infections. J Orthop Res. 2013 Nov;31(11):1694-9. doi: 10.1002/jor.22423.

-        Drago L, et al. Dithiotreitol pre-treatment of synovial fluid samples improves microbiological counts in peri-prosthetic joint infection. Int Orthop. 2023 May;47(5):1147-1152. doi: 10.1007/s00264-023-05714-z.

-        Oliva A, et al. Challenges in the Microbiological Diagnosis of Implant-Associated Infections: A Summary of the Current Knowledge. Front Microbiol. 2021 Oct 29;12:750460. doi: 10.3389/fmicb.2021.750460.

-        Di Domenico EG, et al. The Current Knowledge on the Pathogenesis of Tissue and Medical Device-Related Biofilm Infections. Microorganisms. 2022 Jun 21;10(7):1259. doi: 10.3390/microorganisms10071259.

-        Portillo ME, et al. Improved diagnosis of orthopedic implant-associated infection by inoculation of sonication fluid into blood culture bottles. J Clin Microbiol. 2015 May;53(5):1622-7. doi: 10.1128/JCM.03683-14.

-        Watanabe S, et al. Differences in Diagnostic Sensitivity of Cultures Between Sample Types in Periprosthetic Joint Infections: A Systematic Review and Meta-Analysis. J Arthroplasty. 2024 Mar 13:S0883-5403(24)00232-8. doi: 10.1016/j.arth.2024.03.016. Epub ahead of print. PMID: 38490568.

-        Li C, et al. Meta-analysis of sonicate fluid in blood culture bottles for diagnosing periprosthetic joint infection. J Bone Jt Infect. 2018 Dec 24;3(5):273-279. doi: 10.7150/jbji.29731.

Dear reviewer,

Thank you very much for your work in improving our manuscript. We have tried to do everything in our power to implement them as much as possible. Thank you.
